# Role of TRP Channels in Metabolism-Related Diseases

**DOI:** 10.3390/ijms25020692

**Published:** 2024-01-05

**Authors:** Fengming Wu, Siyuan Bu, Hongmei Wang

**Affiliations:** School of Medicine, Southeast University, Nanjing 210009, China; 213190486@seu.edu.cn (F.W.); siyuanbu@126.com (S.B.)

**Keywords:** TRP channel, diabetes, hypertension, atherosclerosis, nonalcoholic fatty liver disease, oxidative stress

## Abstract

Metabolic syndrome (MetS), with its high prevalence and significant impact on cardiovascular disease, poses a substantial threat to human health. The early identification of pathological abnormalities related to MetS and prevention of the risk of associated diseases is of paramount importance. Transient Receptor Potential (TRP) channels, a type of nonselective cation channel, are expressed in a variety of tissues and have been implicated in the onset and progression of numerous metabolism-related diseases. This study aims to review and discuss the expression and function of TRP channels in metabolism-related tissues and blood vessels, and to elucidate the interactions and mechanisms between TRP channels and metabolism-related diseases. A comprehensive literature search was conducted using keywords such as TRP channels, metabolic syndrome, pancreas, liver, oxidative stress, diabetes, hypertension, and atherosclerosis across various academic databases including PubMed, Google Scholar, Elsevier, Web of Science, and CNKI. Our review of the current research suggests that TRP channels may be involved in the development of metabolism-related diseases by regulating insulin secretion and release, lipid metabolism, vascular functional activity, oxidative stress, and inflammatory response. TRP channels, as nonselective cation channels, play pivotal roles in sensing various intra- and extracellular stimuli and regulating ion homeostasis by osmosis. They present potential new targets for the diagnosis or treatment of metabolism-related diseases.

## 1. Introduction

Metabolic syndrome (MetS) has become a global concern, no longer exclusive to European and American countries, due to the advancement of the social economy and the proliferation of Western lifestyles. The incidence rate of MetS observed within urban populations of certain developing nations frequently surpasses that in Western countries [1]. MetS encompasses various factors such as obesity, insulin resistance, impaired glucose tolerance, hypertension, dyslipidemia, and others [1,2,3]. The risk factors comprising MetS confer risks for cardiovascular disease and Type 2 Diabetes Mellitus (T2DM), with their co-occurrence exacerbating the incidence and severity of cardiovascular disease, thereby posing a significant threat to human health [4,5]. Therefore, early identification of pathological abnormalities related to MetS and proactive prevention of the risk of related diseases is essential.

Transient receptor potential channel (TRP) is a kind of nonselective cation channel on the cell membrane, including seven families: TRPC (canonical), TRPV (vanilloid), TRPM (melastatin), TRPA (ankyrin), TRPP (polycystin), TRPML (mucolipin), and TRPN (drosophila NOMPC). And TRPY was just found and named in yeast [6]. In recent years, research has shown that TRP channels play a significant role in various life activities, such as by sensing different intracellular and extracellular stimuli and maintaining ion homeostasis. TRP channels are expressed in various tissues, including the skeletal muscle, pancreas, liver, kidney, vascular endothelium, adipose tissue, and others. They have been linked to metabolic diseases such as obesity, diabetes, dyslipidemia, atherosclerosis, hypertension, and non-alcoholic fatty liver disease [7,8]. At present, numerous modulators of TRP channels have been documented to exert significant roles in the pathogenesis of metabolic diseases (Table 1). They exhibit a unique role in oxidative stress. This review will provide a comprehensive overview of the impact of TRP channels on metabolic diseases, aiming to provide new directions and ideas for the diagnosis and treatment of metabolic diseases.

### TRP Channel Overview

In 1969, Cosens and Manning [9] discovered that a mutant strain of Drosophila melanogaster, known as TRP (short receiver potential channel), exhibited a transient response to light stimulation that differed from the sustained, platform-like retinal electrical changes observed in the wild-type strain. TRP channels are expressed extensively across various tissues, cells, and organelles, and are involved in a range of physiological and pathological processes, sensing internal and external stimuli, and regulating ion homeostasis and membrane voltage. TRP channels exhibit a wide range of ends and domains, with the sole structural feature common to all TRP structures being the presence of two transmembrane domains and amphiphilic TRP helices. Specifically, the TRP channel comprises six transmembrane helical domains, with both C and N termini located within the cell. The transmembrane domain encompasses the voltage-sensing-like domain (VSLD), consisting of the S1–S4 domain, and the pore domain (PD), formed by the S5–S6 transmembrane helix, which facilitates ion conduction. Following the S6 transmembrane spiral, the amphiphilic TRP spiral is arranged along the inner plane of the cell membrane [10].

Most TRP channels can respond to various types of intracellular and extracellular stimuli, such as light, voltage, sound, temperature, mechanical stimulation, pH, osmotic pressure, chemicals, and cytokines; TRP channels integrate a variety of chemical and physical stimuli. Most TRP channels are nonselective cation channels, allowing Na^+^, Ca^2+^, Mg^2+^, and K^+^ to penetrate through cell membranes, and their gating mechanism is multi-mode, including voltage, temperature, and ligand gating channels [11]. It has been proven that protein kinases A, C, and G (PKA, PKC, and PKG, respectively) and calmodulin can regulate the activity of TRP channels. In addition, some studies have shown that all mammalian TRPC channels require PLC for activation. However, there are differences in their respective selectivity for cations such as Ca^2+^ and in the coupling mechanisms linking PLC to channel activation. The TRP channel is of great significance in sensory physiology. When the electrical signal generated through the TRP channel reaches the stimulation threshold, nerve discharges will be generated and impulses will be transmitted to the corresponding areas of the brain, leading to the perception of stimulation. This is important for animals’ perception of the outside world (such as through touch, hearing, taste, smell, vision, and temperature) and allows single cells to perceive and respond to local environmental changes. In addition, transmembrane cations mediated by TRP channels increase the concentration of the intracellular Na^+^ and Ca^2+^ ions and cause depolarization, achieving the regulation of intracellular Na^+^, Ca^2+^ ions, and membrane voltage and also having a role in the function of movement, such as through controlling muscle contraction and vascular contraction and diastoles [12,13,14].

**Table 1 ijms-25-00692-t001:** Expression of TRP channels and their modulators in metabolism-related diseases.

TRP Channel	Tissue Distribution	Agonist (Endogenous Modulators)	Agonists (Exogenous Regulator)	Antagonists (Exogenous Regulator)	Refs.
TRPC1	pancreas, vascular endothelium, liver	LPS, BMP4	(−)-englerin A, tonantzitlolone	2-APB, Cd^2+^, pico145, difluoromethylornithine, chlorogenic acid, TPT	[15,16,17,18]
TRPC3	pancreas, vascular endothelium	DAG, ROS	Diacylglycerols, SAG, OAG, Pyrazolopyrimidines 4n, GSK1702934A, OptoBI-1, OptoDArG, benzimidazole GSK170, artemisinin	Salvianolic acid B, Pyrazole-3, Norgestimate, Progesterone, 2-Anilino-Thiazole Compounds	[15,16,19,20]
TRPC4	pancreas, vascular endothelium	Gi/o protein, LPS, BMP4	plant-derived sesquiterpenoid englerin A, englerin A, (−)-englerin A, tonantzitlolone	pico145, TPT, M084, ML204, NSAIDs	[15,16,17,21,22,23]
TRPC5	vascular endothelium, liver	ROS, NO	plant-derived sesquiterpenoid englerin A, englerin A, riluzole, methylprednisolone, (−)-englerin A, tonantzitlolone, methylprednisolone, Benzothiadiazine derivative, Riluzole, Rosiglitazone	pico145, clemizole hydrochloride, M084, ML204, 2-APB, 2-Anilino-Thiazole Compounds, NSAIDs	[15,16,17,21,22,23]
TRPC6	pancreas, vascular endothelium, liver	DAG, PIP2, TGFβ1, LPS, BMP4, 20-hydroxyeicosatetraenoic acid	SAG, OAG, Pyrazolopyrimidines 4n, GSK1702934A, OptoBI-1, OptoDArG, Hyperforin (IDN5522), benzimidazole GSK170	BI 749327, Salvianolic acid B, SKF-96365, econazole, W7, compound 8009-5364, norgestimate, sildenafil, STS, TPT, Norgestimate	[8,15,19,20]
TRPC7	vascular endothelium	DAG, PIP2, ATP	Pyrazolopyrimidines 4n, OptoBI-1, PPZ1, PPZ2, benzimidazole GSK170	La^3+^, Gd^3+^, SKF96365, 2-APB, Cilostazol	[15,19,24]
TRPV1	pancreas, vascular endothelium, liver	Anandamide, HPETE, HETE, leukotriene B4, NADA, 2-arachidonylglycerol, HODEs	Capsaicin, resiniferatoxin, N-(3-methoxy-4-hydroxybenzyl) oleamide (NE19550), MDR-652, zoledronic acid, Capsaicin, capsaicin, gingerol, H2S, tarantula spider-derived vanillotoxins, evodiamine, Camphor, Eugenol, CBD, 2-APB, DPBA	HC-030031, GDC-0334, GRC-6211, PAC-14028, currently, Capsazepine, Osthole, Monanchomycalin B, Pulchranin A, Pulchranin B, Pulchranin C, DPTHF	[8,23,25,26,27,28,29,30]
TRPV2	pancreas, vascular endothelium, liver	_	Heat (≥52 °C), CBD, CBN, probenecid, IGF-1, THC, THCV, INS, Δ9-THC, PDGF, NHA, LPC, LPI, 2-APB, fMLP, DPBA	SKF96365, tranilast, amiloride, Gd^3+^, RR, SET2, LEA, AEA, Monanchomycalin B, DPTHF	[29,31,32,33,34]
TRPV3	pancreas, vascular endothelium	FPP, NO	Drofenine, Camphor, IA, Serratol, (+)-Borneol, Menthol, Carvacrol, Eugenol, Citral, 6-tert-Butyl-m-cresol, Thymol, Dihydrocarve-ol, (−)-Carveol, THCV, CBD, 2-APB, DPBA, Drofenine	forsythoside B, Citrusinine II, Osthole, Isochlorogenic acid A, Isochlorogenic acid B, Monanchomycalin B, Pulchranin A, Pulchranin B, Pulchranin C, DPTHF, PC5, Bupivacaine, Mepivacaine, Lidocaine, Ropivacaine, Dyclonine	[8,29]
TRPV4	pancreas, vascular endothelium, liver	endocannabinoid anandamide, arachidonic acid, 5,6-EETs, 14,15-EETs, mechanical flow stimuli	Vildagliptin, GSK1016790A, Cannabinoids and Cannabis Extracts, 4α-PDD, THCV, CBD, DPBA, GSK1016790A, bisandrographolide, RN-1747, phorbol ester, apigenin, eugenol, morin, curcumin, hydroxysafflor yellow A, omega-3 fatty acid, puerarin	HC-067047, RN-1734, GSK2798745, GSK2193874, RR, Capsazepine, Citral, Piperidine-Benzimidazole	[8,23,28,29,35,36]
TRPV5	pancreas	PI(4,5)P2	_	ZINC17988990, ZINC9155420, Econazole, CaM, Gentamicin	[37,38,39,40,41]
TRPV6	pancreas, liver	PI(4,5)P2	Vitamin D, bicalutamide, Capsaicin	CaM, 2-APB, econazole, PCHPD, natural phytoestrogen genistein	[37,40]
TRPM1	retina ON bipolar cells, skin melanocytes	Pregnenolone sulfate	_	Zn^2+^, Intracellular divalent cations	[42,43]
TRPM2	pancreas, liver	ROS, ADP, cADPR, 2′-P-ADPR, 3′-P-ADPR, 2-F-ADP, AMPCPR, Ca^2+^	TRPM2-S, H_2_O_2_, Se, Docetaxel, 5-Fu, LCV	JNJ-28583113, Methotrexate, econazole, clotrimazole, flufenamic acid, N-(p-amylcinnamoyl)anthranilic acid, 2-APB, Scalardial, 3-MFA, icilin, WS-12, 8-Br-cADPR, 8-Br-ADPR, 8-Ph-ADPR, 8-Ph-2′-deoxy-ADPR, 8-(3-acetylphenyl)-ADPR, 8-thiophenyl-ADPR, AMTB, TC-I 2014, DVT, Tricostatin A, sodium butyrate, CTZ, FFA, JNJ-28583113, tat-M2NX	[23,43,44,45,46,47,48]
TRPM3	pancreas	pregnenolone sulfate, β-cyclodextrin, epiallopregnanolone sulphate, sphingosine-1, nifedipine	CIM0216, Nifedipine	17β-estradiol	[42,43,44,48,49,50,51,52]
TRPM4	pancreas, vascular endothelium	Ca^2+^, ATP, calmodulin, IP3, protein kinase C-dependent phosphorylation	voltage-modulated, heat (15–35 °C)	adenine nucleotides, ATP, ADP, AMP, DVT, 9-Phenanthrol	[12,43,48,53]
TRPM5	pancreas	Ca^2+^, PIP2, Steviol glycosides, Rutamarin, glucose	voltage-modulated	TPPO, triphenylphoshine oxide	[12,43,46,53]
TRPM6	pancreas	Mg^2+^, PIP2	2-APB	RR, GTPγS, Gq linked receptor	[43,54]
TRPM7	pancreas, liver	ROS, breakdown of PIP2	Clozapine, Naltriben, proadifen, doxepin, A3 hydrochloride, mibefradil, U-73343, CGP-74514A, metergoline, L-733,060, A-77636, ST-148, clemastine, desipramine, sertraline, methiothepin, NNC 55–0396, prochlorperazine, nortriptyline	2-APB, spermine, MnTBAP, waixenicin A, TG100-115	[23,43,55,56,57]
TRPM8	pancreas, vascular endothelium, liver	Ca^2+^	Menthol, menthoxypropanediol, Camphor, (+)-Borneol, Menthol, Thymol, cilin, eucalyptol, agonist (cold), calcium, AITC, pH modulated	RQ-00434739, KRP-2529, M8-B hydrochloride, Carvacrol, 2-APB, WS-12, CPS-369, AMTB, TC-I 2014, BCTC, Clotrimazole, DD01050, JNJ41876666, RQ-00203078	[8,12,29,43,58,59,60,61]
TRPA1	pancreas, vascular endothelium	catechol estrogen	cinnamaldehyde, cuminaldehyde, AS1269574, JT010, dibenz, capsaicin, Qutenza, GNE551, Cannabinoids and Cannabis Extracts, Carvacrol, Eugenol, Thymol, CBD	Curcumin derivative J147, GRC17536, A-967079, HC-030031, Camphor, (+)-Borneol, Pulchranin A, Pulchranin B, Pulchranin C	[19,20,23,29,62,63,64]

Note: 12- and 15-(S)-hydroperoxyeicosatetraenoic acids (HPETE), 1-oleoyl-2-acetyl-sn-glycerol (OAG), 1-stearoyl-2-arachidonyl-sn-glycerol (SAG), 2,2-diphenyltetrahydro-furan (DPTHF), 2-aminoethoxydiphenyl borate (2-APB), 5- and 15-(S)-hydroxyeicosatetraenoic acids (HETE), ADP-ribose (ADPR), allyl isothiocyanate (AITC), arachidonoyl ethanolamide (AEA), bone morphogenetic protein 4 (BMP4), Cannabidiol (CBD), cannabinol (CBN), clotrimazole (CTZ), diacylglycerols (DAG), Diphenylborinic anhydride (DPBA), flufenamic acid (FFA), fMet-Leu-Phe (fMLP), gadolinium (Gd^3+^), hydrogen sulfide (H2S), hydroxyoc-tadecadienoic acids (HODEs), Incensole acetate (IA), insulin-like growth factor-1 (IGF-1), linoleoyl ethanolamide (LEA), Lipopolysaccharide (LPS), lysophosphatidylcholine (LPC), lysophosphatidylinositol (LPI), N-arachidonoyl-dopamine (NADA), neuropeptide head activator (NHA), Nonsteroidal anti-inflammatory drugs (NSAIDs), N-(Furan-2-ylmethyl)-3-((4-(N7-methyl-N′-propylamino)-6-(trifluoromethyl)-pyrimidine-2-yl)thio)-propanamide (SET2), phorbol ester 4α-phorbol 12,13-didecanoate (4α-PDD), phosphatidylinositol 4-phosphate 2 (PIP2), platelet-derived growth factor (PDGF), ruthenium red (RR), Sodium tanshinone IIA sulfonate (STS), Transforming growth factor beta 1 (TGFβ1), Topotecan (TPT), (−)-trans-Δ9-tetrahydrocannabidol (THC), Δ9-tetrahydro-cannabivarin (THCV).

## 2. TRP Channels: Roles in Oxidative Stress and Organ-Specific Functions

### 2.1. Role of TRP Channels in Oxidative Stress

Oxidative stress, an imbalance between the production of reactive oxygen species (ROS) and the antioxidant defense system, can lead to the oxidation of proteins with redox-sensitive amino acids, such as methionine and cysteine, and oxidative damage to DNA, lipid, and protein. The accumulation of ROS in cells has been associated with various diseases, including diabetes, atherosclerosis, hypertension, ischemia-reperfusion injury, cancer, and neurodegenerative diseases. ROS can also act as a signaling molecule to regulate vascular tension and immune function. Some TRP channels, including TRPM2, TRPM7, TRPC5, TRPV1, and TRPA1, act as sensors for cellular redox state changes and can be activated by hydrogen peroxide (H_2_O_2_), nitric oxide (NO), and electrophilic reagents to trigger appropriate cellular responses to environmental redox stimuli. Oxidative stress refers to the imbalance between the production of ROS/RNS and the antioxidant defense system. The production of ROS/RNS induces the oxidation of proteins with redox-sensitive amino acids (such as methionine and cysteine). The oxidative damage to DNA, lipids, and proteins is mediated by protein oxidation and lipid peroxidation, leading to cell dysfunction. Some studies have shown that the accumulation of ROS in cells may be related to diabetes, atherosclerosis, hypertension, ischemia-reperfusion injury, cancer, neurodegenerative diseases, and other diseases [65,66,67]. In addition, ROS may act as a molecule mediating cell signal response to regulate vascular tension and immune function [68,69]. Endogenous pathological lipids produced under oxidative stress, such as LPO metabolite nitro fatty acid, can activate TPRA1 through covalent binding. Some metabolites produced by lipoxygenase (LOX) catalysis can be used as activators of TRPV1 [70]. The lipid peroxidation of polyunsaturated fatty acids produces endogenous active aldehydes that can covalently modify and activate TRP channels, leading to calcium influx and cytokine release [71].

TRP channels may be involved in the occurrence and development of metabolic diseases by connecting with metabolic tissues (liver, pancreas, skeletal muscle, and so on) as redox sensors, inflammatory regulators, and modulators of hormone secretion [72]. Some TRP channels act as sensors for cellular redox state changes, such as TRPM2, TRPM7, TRPC5, TRPV1, and TRPA1. They can be activated by hydrogen peroxide (H_2_O_2_), nitric oxide (NO), and electrophilic reagents to trigger appropriate cellular responses (such as cell death, inflammation, cytokine release, ROS detection, etc.) to environmental redox stimuli [73,74,75]. TRPML1 is a ROS sensor located on the lysosomal membrane. ROS activate the lysosomal TRPML1 channel to induce lysosomal Ca^2+^ release, trigger the nuclear translocation of transcription factor EB (TFEB), promote autophagy, remove damaged macromolecules and organelles, and maintain the redox balance of cells (Figure 1) [76].

Oxidants can activate TRPM2 channels either directly, by acting on TRPM2 channels, or by prompting PARP to produce ADPR, the second messenger of oxidative-stress-induced TRPM2 gating [77]. TRPM2 can mediate the activation of ROS-dependent NLRP3; ROS stimulates the production of ADPR, and as a second messenger, ADPR induces Ca^2+^ influx through TRPM2. The increased intracellular Ca^2+^ concentration activates the NLRP3 inflammasome and triggers the inflammatory response (Figure 1) [78].

Early research supports the idea that TRPM2 activation aggravates inflammation and tissue damage through continuously increasing intracellular Ca^2+^ or cytokines. For CNS neurons and β cells in the pancreas, studies on the relationship between the TRPM2 channel and oxidative stress in cells showed that the ROS-activated TRPM2 channel led to changes in intracellular Ca^2+^ and Zn^2+^ homeostasis, mitochondrial dysfunction, and mitochondrial-induced apoptosis. In particular, ROS induces TRPM2 activation to mediate mitochondrial Zn^2+^ accumulation and stimulate the recruitment of dynein-related protein 1 (Drp-1); furthermore, these signals mediate mitochondrial fragmentation and dysfunction, which result in the apoptosis of β cells [79]. On the other hand, recent studies have shown that TRPM2 protects many physiological processes. TRPM2-mediated Ca^2+^ entry enhances the translation and stability of hypoxia-inducible factor-1/2α (HIF-1/2α). HIF-1/2α expression is reduced when TRPM2 is knocked out, and the decrease in mitochondrial autophagy and the increase in ROS can be caused by the reduction of HIF-1/2α (Figure 1). Based on this, the above results reflect the significance of TRPM2 in supporting mitochondrial function and reducing ROS-induced cell damage [80]. The increased sensitivity of the TRPM7 channel to intracellular Mg^2+^ may be involved in the inhibition of TRPM7 by ROS under high [Mg^2+^]i conditions. The inhibitory effect increases with the decrease in [ATP]i, while no obvious inhibition of the TRPM7 current is found at normal [ATP]i [81].

According to reports, TRPV4 has been found to augment oxidative stress by impeding catalase and glutathione peroxidase while elevating the activity of neuronal nitric oxide synthase (nNOS) (Figure 1) [82]. Similarly, TRPA1 has been implicated in mitochondrial calcium influx, mitochondrial ROS production, mitochondrial membrane depolarization, and mitochondrial damage during ATP-induced mitochondrial damage, ultimately resulting in oxidative stress and inflammation [83]. The upregulated TRPA1 expression after traumatic brain injury (TBI) may cause secondary injury through calcium influx, while blocking TRPA1 can alleviate the oxidative-stress-induced injury after TBI through the CaMKII/AKT/ERK signaling pathway [84].

### 2.2. TRP Channels in Organ Tissues: Localization and Functions

Following the discussion on the role of TRP channels in oxidative stress, we will discuss their specific localization and functions in various organ tissues, including the pancreas, liver, and vascular endothelium. These channels have been implicated in various molecular processes related to metabolic diseases, and their study can provide valuable insights for the diagnosis and treatment of these diseases.

#### 2.2.1. Function of TRP Channels in the Pancreas

The secretion of insulin in response to heightened levels of blood glucose is predominantly linked to electrical activity, with the primary mechanism being the ATP/ADP ratio-triggered pathway and Ca^2+^ inflow-induced insulin release. Glucose is able to permeate the cell via a glucose transporter molecule (GLUT), and its metabolism is initiated through glucose phosphorylation catalyzed by glucokinase in the cytoplasm, which is accompanied by an elevation in the ATP/ADP ratio, resulting in the closure of ATP-sensitive K^+^ channels (K^+^_ATP_) and the depolarization of the plasma membrane. When the plasma membrane depolarizes to −50 mV, the voltage-gated calcium channel opens, causing Ca^2+^ inflow and an increase in cytoplasmic calcium ion concentration, and the increase in intracellular Ca^2+^ concentration triggers the exocytosis of insulin-containing vesicles in β cells [85,86]. Depolarization due to the inhibition of the ATP-sensitive potassium channel and inward depolarization current through the transient receptor potential (TRP) channel leads to electrical activity, the opening of the voltage-gated calcium channel, and exocytosis of insulin (Figure 2) [87].

The regulation of insulin secretion in β cells of the pancreas can be achieved through the activation of multiple TRP channels, which can modulate membrane depolarization and calcium influx [88,89]. This process involves the activation of TRP channels via various G protein signal pathways, resulting in an increase in intracellular Ca^2+^ concentration and the reinforcement of whole-cell currents [90]. Additionally, evidence suggests that TRP channel activation can also induce an intracellular flow of Ca^2+^, in addition to the rise in intracellular Ca^2+^ concentration that is dependent on K^+^_ATP_ shutdown [91]. The activation of TRPA1 can cause calcium influx and membrane depolarization in β cells of the rat pancreas to mediate insulin release, even when voltage-gated sodium and calcium channels are blocked and K^+^_ATP_ is activated [92]. Applying catechol estrogen can activate the TRPA1 channel on β cells to mediate the increase in cytosolic free calcium concentration and GSIS, which is inhibited by the specific inhibitors of the TRPA1 channel (A-967079 and HC030031) or TRPA1 siRNA [62]. Another TRP channel, TRPM2, was also found in β cells. Compared with wild-type mice, TRPM2 knockout mice have impaired glucose clearance accompanied by decreased plasma insulin levels, and their glucose stimulation and GLP-1-enhanced insulin secretion are impaired [93]. Some research shows that TRPM2 and insulin (a marker of β cells) are highly co-expressed, which indicates that the TRPM2 may have corresponding functions in pancreatic β cells. The insulin release from β cells is related to the TRPM2 channel, which may involve the activation of the glucagon-like peptide-1 (GLP-1) receptor and its downstream PKA-dependent phosphorylation of TRPM2 or its closely related proteins to enhance the activity of TRPM2 and ultimately realize K^+^_ATP_ non-dependent Ca^2+^ inflow and insulin release [94].

According to reports, the maintenance of structural integrity and signal transduction capacity in β cells is regulated by TRPV1, which is driven by Ca^2+^ and controls the secretagogin-regulating axis. Empirical investigations have demonstrated that the activation of Sp1-dependent promoters can facilitate the expression of secretin mRNA and secretin protein induced by TRPV1, and that increased secretin levels enhance glucose-stimulated insulin secretion (GSIS) in β cells [95]. TRPV1 has been found to exert an impact on the viability of pancreatic β cells via perturbations in protein folding and the dysregulation of USP9X deubiquitination enzyme activity, contingent upon the presence of secretagogin. Notably, extant research has demonstrated that the absence of secretagogin results in the deactivation of ubiquitin carboxyl-terminal hydrolase (USP9X and USP7), thereby elevating the incidence of β cell apoptosis [96]. Furthermore, TRPV1 can be activated by capsaicin and stimulate glycogen synthesis and insulin secretion via the TRPV1-PDX-1-GLUT2/GK and TRPV1-PDX-1-IRS1/2 signaling pathways [63]. Several pertinent studies have demonstrated that TRPV2 augments glucose-stimulated insulin secretion (GSIS) by means of stimulating glucagon-like peptide-1 (GLP-1) secretion. This mechanism is attributed to the binding of lysophosphatidylinositol (LPI) to G protein-coupled receptor 55 (GPR55), which activates Gq and G12/13, leading to TRPV2 activation through actin reorganization. Consequently, TRPV2 participates in the LPI-induced elevation of intracellular Ca^2+^ concentration and GLP-1 secretion in enteroendocrine L cells, while GLP-1, in turn, enhances GSIS in pancreatic β cells. Furthermore, the utilization of the total internal reflection fluorescence (TIRF) microscope facilitated the observation of the localization of TRPV2 on the plasma membrane. The results indicated that the administration of LPI stimulated the translocation of TRPV2 to the plasma membrane via the activation of GPR55. This process subsequently contributed to the elevation of intracellular Ca^2+^ concentration and the secretion of GLP-1 [97].

The following studies show that other members of the TRP family also participate in insulin release. In INS-1E cells, TRPV4 can be activated by stimulation with hypotonic solution and moderate heating, as well as 4α-PDD (TRPV4 activator), while elevating intracellular calcium ions. Among other things, TRPV4 can mediate GSIS enhancement in response to pharmacological activation by 4α-PDD [98]. TRPC can interact with Orai1 to form a functional SOC and induce SOCE (store-operated Ca^2+^ entry) through electrostatic interaction by the polybasic lysine-rich domain of the stormal interaction molecule 1 (STIM1) protein. In the endoplasmic reticulum, when Ca^2+^ depletes, TRPC1, Orai1, and STIM1 form a ternary complex, and the inhibition of the ternary complex will reduce insulin release [99].

#### 2.2.2. Function of TRP Channels in the Liver

##### TRPV Channels

Several members of the TRP channel family have been identified in liver tissues or hepatocyte lines through various channel measurement techniques such as PCR, in situ hybridization, and immunofluorescence [100]. Given the liver’s crucial role in carbohydrate, fat, and protein metabolism, it is imperative to investigate the influence of TRP channels on the liver, particularly in relation to its fundamental functions in the detoxification and regulation of systemic homeostasis.

The investigation into the proteomic alterations in the liver of wild-type mice and TRPV1 knockout mice revealed 18 distinct protein expression variances between the two cohorts of liver proteomics. Of these variances, 17 proteins were upregulated, while one was downregulated in TRPV1 knockout mice. These differential proteins may suggest a correlation between TRPV1 and liver metabolism and dysfunction. The downregulation of Transient Receptor Potential Vanilloid 1 (TRPV1) expression in liver cirrhosis tissue, caused by a variety of etiologies, has been observed [101]. This study demonstrates a significant decrease in TRPV1 expression in liver cirrhosis tissue due to diverse etiologies. Importantly, the silencing of TRPV1 prompts the proliferation and activation of Hepatic Stellate Cells (HSCs) by enhancing Transforming Growth Factor-β (TGF-β) stimulation. This leads to an increase in the production of Extracellular Matrix (ECM) proteins, including α-smooth muscle actin and collagen I (Figure 3). The findings suggest that TRPV1 may have a protective impact on liver fibrosis, while TRPV4 may contribute to liver damage by facilitating acetaminophen (APAP) metabolism [102]. The inhibition of TRPV4 via pharmacological means or genetic deletion may mitigate APAP-induced oxidative stress and mitochondrial membrane depolarization, thereby ameliorating liver cell injury caused by APAP [103].

##### TRPM Channels

TRPM2 exhibits a heightened susceptibility to oxidative stress and serves as a conduit for cell death instigated by both endogenous and exogenous oxidative stressors, which disrupt the equilibrium of intracellular ions. This process may entail the direct stimulation of the TRPM2 channel by reactive oxygen species (ROS) and the activation of TRPM2 via the production and escalation of ADPR, instigated by PARP/PARG in the nucleus or mitochondrial NADase [45]. In rat hepatocytes, the majority of TRPM2 proteins are situated intracellularly and can be translocated and dispersed to the plasma membrane of hepatocytes in response to oxidative stress [104]. The activation of TRPM2 by ROS may be attributed to the elevation of ADP-ribose (ADPR) levels induced by ROS. ADPR, in conjunction with Ca^2+^, can bind to the NUDT9H structural domain of the TRP channel, thereby facilitating ROS-mediated hepatocyte injury via the opening of the TRP channel (Figure 3) [105].

The potential association between TRPM8 and liver regeneration in mice may be mediated by changes in intracellular Ca^2+^. The absence of TRPM8 has been observed to impede liver regeneration, potentially leading to a disruption in the energy metabolism of hepatocyte mitochondria through a decrease in PGC1α signaling (Figure 3) [106]. Additional research has demonstrated that TRPM2, TRPM6, TRPM7, and TRPM8 may impact hepatic ischemia-reperfusion injury, with TRPM2 knockout exhibiting a reduction in such injury through the activation of autophagy and inhibition of autophagy-negative regulation of the NLRP3 inflammasome pathway [107,108].

##### Other Channels

The study on wild-type mice and TRPC5 knockout mice showed that TRPC5 deletion could alleviate liver dyslipidemia and liver injury induced by cholestasis [109]. TRPC6 and TRPV2 may contribute to the progression of hepatocellular carcinoma, which may be related to the stemness of hepatocellular carcinoma, migration, and the invasion of hepatocellular carcinoma cells [110,111].

#### 2.2.3. Function of TRP Channels in the Vascular Endothelium

In arterial smooth muscle cells, membrane depolarization leads to the influx of Ca^2+^ through voltage-gated calcium channels, resulting in increased intracellular calcium concentration and the calcium-mediated contraction of smooth muscle cells. In endothelial cells, it is important to realize the following three endothelium-dependent vasodilation pathways: (1) tissue permeability gas NO produced by endothelial nitric oxide synthase (eNOS), (2) PGI2 produced by cyclooxygenase (COX), and (3) endothelium-derived hyperpolarizing factor (EDHF) dependent on K channels which can be activated by calcium ions, such as the small conductance potassium channels (SK) and the intermediate conductance potassium channels (IK) (Figure 4) [112]. The vascular endothelium expresses various members of the TRP channel family, potentially impacting several physiological processes such as thermoregulation, angiogenesis, thrombosis prevention, oxidative stress in endothelial cells, inflammatory response, regulation of permeability, and control of vascular tension via the modulation of intracellular calcium levels [113,114]. TRP channels can mediate Ca^2+^ influx into endothelial cells and vascular smooth muscle cells, thereby regulating vascular function. Among them, TRPC6, TRPC3, TRPM4, TRPV1, TRPV2, TRPM8, and TRPC1 are expressed in smooth muscle cells of various vessel and are related to vascular contraction contractility. At the same time, TRPV1, TRPV3, TRPV4, TRPA1, TRPC3, and TRPC4 play an important role in the endothelium-dependent vasodilation of some vessel [115]. It has also been reported that TRPC1, TRPC3, TRPC4, TRPC5, TRPC6, TRPV1, TRPV4, TRPM2, TRPM4, TRPM7, TRPA1 and other TRP channel family members participate in angiogenesis [28,116].

##### TRPV Channels

According to reports, TRPV1 has the potential to facilitate UCP2-mediated endothelial dysfunction [117]. The dysregulation of the endothelial TRPV4 channel may be associated with endothelial cell dysfunction and may contribute to the onset and progression of cardiovascular diseases such as diabetes, hypertension, and obesity [118,119]. TRPV4 is involved in the endothelium-dependent vasodilation EDHF pathway in response to blood flow and acetylcholine (ACh) in endothelial cells. TRPV4 generates local calcium signals via Ca^2+^ influx, which subsequently activates intermediate conductance potassium channels (IK) and small conductance potassium channels (SK) with high Ca^2+^ sensitivity. The resultant hyperpolarization current is transmitted to adjacent smooth muscle cells via gap junctions, leading to vasodilation [120,121]. In addition, TRPV4 can also affect endothelial function through the TRPV4/microRNA-6740/ET-1 signal axis [122]. It is reported that TRPC3 can also promote the activation of endothelial SK and IK and mediate endothelium-dependent hyperpolarization (EDH) to achieve vasodilation [123].

TRPV1 activation in EC can trigger Ca^2+^ dependent PI3K/Akt/CaMKII signal transduction, promote the formation of the TRPV1-eNOS complex, and activate eNOS to produce NO, and TRPV1 activation can inhibit endothelial cell inflammation through this pathway [124,125]. Additionally, TRPV1 may modulate endothelial cell function by activating the AMPK signal transduction process [126]. In a separate investigation, TRPV1 was found to decrease superoxide anion production induced by oxidized low-density lipoprotein through the upregulation of peroxisome proliferator-activated receptor α expression in vascular smooth muscle Thus, the proliferation and migration of ox-LDL-induced VSMC were inhibited. In addition, TRPV1 mediated VSMC phenotype conversion and angiogenesis, which may be a valuable target for treating ischemic diseases [127,128].

##### Other Channels

TRPC6 deficiency can regulate vascular tension by reducing the contraction and depolarization of vascular smooth muscle cells (VSMC) caused by arterial pressure. TRPC6 is also involved in the regulation of Akt activity through TGF-β1 and PTEN(phosphatase and tensin homologue deleted from chromosome 10), affecting the differentiation of the VSMC phenotype in the direction of constrictive VSMCs, and thereby affecting the plasticity of VSMC [129,130,131]. In addition, Ang II can upregulate the expression of TRPM7, which is involved in the activation of the Pyk7-ERK2/1-Elk-2 pathway by Ang II; the ERK2/1-Elk-2 signaling pathway can inhibit VSMC differentiation and affect VSMC phenotype switching [132].

#### 2.2.4. Role of TRP Channels in Adipose Tissue

Investigations into TRPC1 have demonstrated that the endogenous entry of Ca^2+^ into both subcutaneous and visceral adipocytes is contingent upon TRPC1-STIM1 [133]. The suppression of TRPC1 in adipocytes can obstruct Ca^2+^ entry and impede adipocyte differentiation [134]. In cultured brown adipocytes, the downregulation of TRPC1 results in the downregulation of UCP1 and PPARγ [135]. TRPC3 can instigate calcium influx, subsequently activating the NF-KB signaling pathway [136]. The administration of the TRPC4/TRPC5 blocker ML204 intensifies high blood glucose, dyslipidemia, fat tissue deposition, hepatic steatosis, and TNFα in HS-fed mice [137]. The dominant negative ion pore mutant of TRPC5 (DNT5) escalates adiponectin transcriptional expression while diminishing the transcriptional expression of the inflammatory mediator Tnfα, and potentially reducing Il6, Il1β, and Ccl2 [138]. In the case of TRPC6, Klotho can preserve TRPC6 in the endoplasmic reticulum and open it to increase reticular Ca^2+^ leakage [139]. TRPC7-mediated Ca^2+^ signaling relies on the AKT and MAPK signaling pathways to inhibit cell cycle progression and cell migration [140]. The overexpression of TRPV1 augments the expression of PPARγ and other thermogenic genes in cultured 3T3-L1 preadipocytes [141]. TRPV2, expressed in brown adipocytes, aids in differentiation and/or thermogenesis. Sensory neurons that express TRPV1 foster increased energy expenditure by stimulating the sympathetic nervous system and the secretion of adrenaline. Research has shown that treatment with olanzapine also boosts the expression of TRPV1/TRPV3 in the nucleus accumbens (NAc) and TRPV3 in the ventral tegmental area (VTA) [142]. The activation of TRPV4 results in the swift phosphorylation of ERK1/2 and JNK1/2, further inhibiting the expression of thermogenic genes, and TRPV4 negatively regulates the expression of PGC1α, UCP1, and cellular respiration [141,143]. TRPV5 and TRPV6 hold a significant role in maintaining high blood Ca^2+^ levels in higher organisms [144]. TRPV5 plays a pivotal role in determining urinary Ca^2+^ excretion levels, but the physiological role of TRPV6 extends beyond intestinal Ca^2+^ absorption. TRPM1 is a recognized tumor suppressor and is considered to be a Ca^2+^-permeable ion channel [145,146]. The growth hormone secretagogue receptor (GHSR) couples uniquely to cAMP/TRP2 signaling in β cells, and this β cell GHSR with unique insulin-stabilizing signaling largely explains the systemic effects of ghrelin on circulating glucose and insulin levels. TRPM3 is an ion channel inhibited by G protein-coupled receptors (GPCRs) through direct interaction with the released G protein (Gβγ) during activation [147]. TRPM4 is a crucial regulator for histamine-induced Ca^2+^ signaling in hASCs and is necessary for adipogenesis [148]. Studies have observed significantly decreased mRNA expression levels of Trpv4, Trpm4, Trpm5, and Trpm7 in differentiated adipocytes. The expression of TRPM6 remains unaffected by 3 mM magnesium and 0.1 mM magnesium under 25 and 250 pM insulin, respectively, but the expression of NIPA1 is decreased. The pro-inflammatory effect of TRPM7 is dependent on Ca^2+^ signaling [149]. TRPM7-initiated Ca^2+^ influx enhances the activation of transforming growth factor-β-activated kinase 1 through the co-regulation of calcium/calmodulin-dependent protein kinase II and tumor necrosis factor receptor-associated factor 6, leading to intensified nuclear factor κB signaling [150]. The activation of TRPM8 in adipocytes leads to Ca^2+^ influx and the activation of protein kinase A (PKA), inducing mitochondrial elongation, the localization of mitochondria to lipid droplets, lipolysis, β-oxidation, and UCP1 expression [151]. This suggests that TRP channels are instrumental to the functionality of adipose tissue. The exploration of the mechanisms underlying TRP channels in adipose tissue is of substantial importance for the prophylaxis and management of cardiovascular diseases.

## 3. TRP Channel and Metabolic Diseases

### 3.1. TRP Channel and Diabetes

Diabetes is a metabolic disorder that is typified by chronic hyperglycemia resulting from defects in insulin secretion and/or insulin utilization. Prolonged disturbances in the carbohydrate metabolism may be secondary to microangiopathy and neuropathy, which can lead to a range of complications such as diabetic nephropathy, diabetic retinopathy, diabetic peripheral neuropathy, atherosclerotic cardiovascular disease, diabetic foot disease, and other conditions that significantly impair the quality of life of affected individuals. TRP channel genes, along with their isoforms, are found in high quantities within human pancreatic α and β cells, serving a multitude of essential functions [152]. Studies indicate that the stimulation of bilateral ST25 acupoints via electroacupuncture (EA) can potentially restore β cell functionality in T2DM rats [153]. This restoration is achieved through the regulation of the TRPV1 channel (SP/CGRP) and insulin circuit. The chemical ablation of TRPV1 neurons under physiological conditions can influence the functionality and quantity of pancreatic β cells, subsequently improving glucose metabolism and positioning β cells as a primary target of TRPV1 neurons [154]. Research has indicated that TRPV1 and TRPA1 are involved in the regulation of glucose homeostasis and are associated with weight management, pancreatic function, hormone secretion, thermogenesis, and neuronal function [155]. Typically, TRPV1, acting as a sensory transmitter, assumes a pivotal role in the detection of external stimuli and can be stimulated by capsaicin (CAP). In individuals with diabetes, the TRPV1 channel may contribute to the sensory disturbances of diabetic peripheral neuropathy, which in turn drives diabetic neuropathic pain [26]. Lee et al. demonstrated that high-fat feeding resulted in increased obesity and insulin resistance in TRPV1 knockout mice compared to wild-type mice [156]. Conversely, Lam et al. observed no significant difference in TRPV1 expression in DRG neurons cultured under normal or hyperglycemic conditions. However, the inhibition of PKCβ and Src kinase, respectively, led to the inhibition of high-glucose-enhanced capsaicin-induced current enhancement [157]. The basal CAP-induced current of the control group had no significant effect. This suggests that in the early stage of diabetes, DRG neurons are affected by high glucose and enhance TRPV1-mediated current through PKC and Src kinase signal transduction. These molecular and electrophysiological changes may be involved in the early sensory abnormalities of diabetes. Bestall et al. [158] show that the TRPV1 activity induced by capsaicin may be increased through the HMGB1-RAGE-dependent mechanism under high glucose, thereby participating in the signal transduction of sensory neurons. The microvascular response mediated by the TRPV1 channel is weakened in T1DM patients, and TRPV1 may participate in the dysfunction of neurovascular microcirculation [159]. In addition, some studies have shown that the inhibition of TRPV1 may improve diabetes-induced endothelial dysfunction and induce vascular regeneration in diabetic mice, and may improve insulin resistance [160,161]. Interestingly, another study indicates that CAP can improve the endothelial dysfunction of diabetes by activating TRPV1/eNOS to reduce oxidative stress and increase NO in vascular endothelial cells of hyperglycemic mice [162].

The TRPV2 channel is known to mediate insulin secretion resulting from the swelling of mouse pancreatic β cells [163]. TRPV4 contributes to the proliferation of pancreatic β cells and insulin production, and its activation can trigger insulin secretion [164]. Gao et al. [35] used streptozotocin-induced type 1 diabetic mice as the experimental group object; after intervention with vildagliptin (DPP-4 inhibitor), the results showed that the EDR of T1DM mice was significantly improved compared with the control group mice. This effect could be blocked by the nitric oxide chelator but not by the GLP-1 receptor inhibitor. This shows that vildagliptin can prevent endothelial dysfunction induced by hyperglycemia through a GLP-1 independent mechanism, and subsequent relevant experiments have further provided the molecular mechanism involved in this process. Vigeltin has been observed to activate TRPV4 through the mechanism of hydrogen bonding, which subsequently promotes calcium uptake in endothelial cells and mitigates the production of endothelial reactive oxygen species (ROS). It is hypothesized that in the context of pancreatic beta cells, TRPV4 may also be modulated by ROS, thereby influencing the secretion of insulin. In INS-1E cells, the influx of Ca^2+^ through the TRPV6 channel can control insulin gene expression, cell viability, and cell proliferation [165].

The inhibition or removal of TRPM2 in mouse islets, along with the silencing of TRPM2 expression in human islets through RNA interference, can prevent FFA/cytokine-induced β cell death [166]. The TRPM3 channel within β cells is critical in managing glucose-dependent insulin release [167]. TRPM4 and TRPM5 are also considered key regulators of glucose-induced insulin secretion in pancreatic β cells. Enhancing TRPM4 and TRPM5 can stimulate a higher insulin secretion from β cells, thereby preventing T2DM onset [168]. TRPM5 is essential for taste signals such as sweet, bitter, and fresh tastes in type II taste receptor cells and mediates glucose-induced insulin secretion in pancreas β cells [168,169]. TRPM5 participates in carbohydrate metabolism and glucose homeostasis regulation by regulating the taste response and insulin secretion, which may play a role in controlling calorie intake and preventing diabetes. Steviol glycoside could enhance GSIS in islets of WT mice but not in TRPM5 knockout mice [170]. TRPM7, one of the most abundantly expressed TRP channels in human β cells, can govern pancreatic development and β cell proliferation in transgenic mouse models by maintaining Mg^2+^ homeostasis [171,172]. Moreover, the elimination of TRPM7 in INS-1 cells can augment the insulin response to glucose [173]. TRPC1 can contribute to insulin secretion in rat pancreatic β cells by mediating the functionality of store-operated Ca^2+^ (SOC) channels [174]. TRPC3, functionally expressed in human and mouse pancreatic islet cells, can lead to defective insulin secretion and impaired glucose tolerance in pancreatic β cells if pharmacologically inhibited or removed [175]. TRPC4 is suggested to be involved in M2 and M3 muscarinic receptor signaling, implying its potential role in acetylcholine-induced insulin action in native pancreatic β cells [176].

### 3.2. TRP Channel and Hypertension

Hypertension is a clinical syndrome that is distinguished by elevated systemic arterial pressure, which is influenced by both genetic and environmental factors. The pathogenesis of hypertension is associated with alterations in vascular structure and tension, the activation of the sympathetic nervous system, the activation of the renin–angiotensin–aldosterone system, and insulin resistance. Hypertension is a notable risk factor for cardiovascular disease, and it can lead to severe complications such as cerebrovascular disease, coronary heart disease, aortic dissection, and chronic renal failure, thereby posing a significant threat to human life and health. There is a growing body of evidence in the literature highlighting the involvement of TRP channels, including TRPM7/8, TRPV1, and TRPA1, in the pathophysiology of hypertension.

#### 3.2.1. TRPM Channels

Various members of the TRP channel family modulate vascular tension and blood pressure by regulating intracellular calcium homeostasis and cellular calcium signal transduction, thereby influencing vascular contraction and relaxation. Disordered functions of these channels may lead to vascular dysfunction and hypertension [177,178]. Through an examination of the mouse transcriptome database, TRPM7, the most prevalent TRP superfamily gene in the carotid body, was identified as a potential candidate for the nonselective blocking of the TRP family to eliminate leptin-induced carotid sinus nerve afferent activity and the subsequent elevation of blood pressure. By increasing the methylation of the TRPM7 promoter, leptin receptor-deficient mice and leptin-deficient mice can reduce Trpm7 transcription and expression, demonstrating how the TRPM7 channel contributes to leptin-induced hypertension. Furthermore, after the local application of TRPM7 blocker FTY720 or TRPM7 shRNA in CB, it was found that it inhibited leptin-induced hypertension effectively [179,180].

Under physiological conditions, there is a balance between salt taste preference caused by epithelial sodium channels and salt taste aversion caused by TRPM5 in the tongue epithelium. However, a long-term high-salt diet can downregulate TRPM5 or inhibit PKC-dependent threonine phosphorylation, which impairs TRPM5-mediated high salt aversion without affecting epithelial sodium channel-dependent salt taste preference, suggesting that the inhibition or deletion of TRPM5 as a taste sensor may be involved in high salt-induced hypertension [181]. TRPM8, as a cold sensor, may alleviate cold-induced hypertension related to the renin–angiotensin–aldosterone system by inhibiting Ang II-induced cytoplasmic Ca^2+^influx and mitochondrial calcium overload after being activated by menthol [182].

#### 3.2.2. TRPV Channels

Ottolini et al. [183] show that AKAP150-PKC-TRPV4 signal transduction increases the activity of the TRPV4 channel in endothelial cells to promote endothelium-dependent vasodilation and help regulate resting blood pressure. Mice that specifically knock out endothelium-specific TRPV4 and AKAP150 show elevated resting blood pressure. Under physiological conditions, TRPV4 SK/IK pathway induces endothelium-dependent hyperpolarization (EDH) through gap-junction-mediated hyperpolarization current conduction between endothelial cells and surrounding smooth muscle cells. In addition, the TRPV4-eNOS pathway also participates in regulating vascular relaxation. However, in the process of hypertension, the expression of TRPV4 and SK protein in the endothelium was significantly reduced, and the impairment of the TRPV4-eNOS pathway may reveal that endothelium TRPV4 participates in the pathophysiological changes in hypertension [184,185].

TRPV1 plays a role in the neural modulation of blood pressure, and its receptor agonist, resiniferatoxin (RTX), is a specific afferent neurotoxin. The application of RTX to the T1–T4 DRG epidural region can effectively eliminate the cardiac sympathetic afferent reflex (CSAR). Shanks et al. performed an epidural drug intervention targeting the T1–T4 chest region in 8-week-old spontaneously hypertensive rats (SHR). The RTX intervention group exhibited blood pressure levels that remained in close proximity to the baseline, whereas the carrier control group experienced an increase of 20–25 mmHg in mean arterial blood pressure. Upon administering treatment to 16-week-old SHR rats, the findings indicated that the RTX intervention group elicited a significant reduction in mean arterial pressure when compared to the carrier control group [27]. Furthermore, the thermosensitive receptors, TRPA1/TRPM8/TRPV1, which are expressed in the cardiovascular system, have been implicated in the pathogenesis of hypertension [186]. The distinct activation and deactivation processes of these channels suggest a potential signaling pathway that could pave the way for the development of novel therapeutic strategies for hypertension and associated vascular disorders in the future.

### 3.3. TRP Channel and Atherosclerosis

Atherosclerosis (AS) is a pathophysiological phenomenon that triggers a series of intricate events in the arterial wall, influenced by risk factors such as diabetes, hypertension, dyslipidemia, obesity, smoking, and others. The development of the lesion primarily involves the infiltration of LDL-C into the intima through the damaged endothelium, which undergoes oxidative modification to ox LDL-C. Subsequently, macrophages and smooth muscle cells proliferate and migrate to the subintima under the influence of adhesion factors and cytokines, where they recognize and engulf ox LDL-C and transform into foam cells, ultimately leading to the formation of lipid stripes. Wei et al. conducted an experiment wherein ApoE knockout mice were administered evodiamine and carrier, and the sizes of the atherosclerotic lesions and lipid metabolism parameters were measured [187]. The findings revealed that the atherosclerotic lesions were significantly smaller after evodiamine intervention, and the total cholesterol and non-HDL-c levels were reduced. The results of the study suggest that the anti-atherosclerotic effect of evodiamine may be dependent on TRPV1, as evidenced by the higher size of atherosclerotic lesions and serum levels of lipid mass spectrometry observed in ApoE/TRPV1 double-knockout mice treated with evodiamine compared to single ApoE knockout mice. The activation of TRPV1 may be linked to the upregulation of ABCA1 expression and downregulation of low-density lipoprotein receptor-related protein 1 (LRP1) expression in vascular smooth muscle cells (VSMCs), potentially modulating lipid metabolism and mitigating lipid accumulation by promoting cholesterol efflux and inhibiting uptake [25]. Gao et al. [188] prepared the CuS-TRPV1 switch by coupling CuS NP with TRPV1 monoclonal antibody. Following NIR irradiation, the activation of the TRPV1 channel resulted in an increase in intracellular calcium concentration, the activation of the AMPK-dependent autophagy pathway, and an increase in cholesterol efflux, thereby reducing lipid accumulation and foam cell formation in VSMC. These findings suggest that the photothermal-activated CuS-TRPV1 switch may mitigate atherosclerosis through the aforementioned molecular or cellular mechanisms. Additionally, capsaicin was observed to induce autophagy via the autophagy–lysosome pathway and AMPK-dependent autophagy pathway by activating TRPV1, thereby impeding the formation of VSMC foam cells [189]. Furthermore, it is plausible that TRPV4 plays a role in the modulation of macrophage foam cell formation by regulating the uptake of ox-LDL [190].

Additionally, TRPA1 has the potential to influence the phenotypic transition between macrophages M1 and M2 at the epigenetic level. The suppression of TRPA1 may regulate the differentiation of macrophages towards the M1 inflammatory phenotype and consequently impede the progression of atherosclerosis [191]. Moreover, TRPA1 may be implicated in cholesterol metabolism and inflammatory disorders, thereby contributing to the development of atherosclerosis. Zhao et al.’s research demonstrated that the expression of TRPA1 was elevated in macrophages located in the aortic atherosclerotic plaque of ApoE knockout mice. Upon the administration of HC030031 to suppress TRPA1 activity, serum lipid mass spectrometry indicated an increase in total serum cholesterol, non-HDL-c, and triglyceride levels [192]. The potential mechanism at play may entail the knockout of TRPA1 or the inhibition of its activity through HC030031, which could impede the reverse efflux of cholesterol by diminishing the expression of ABCA1 and ABCG1, ultimately leading to the accumulation of lipids in macrophages. Upon stimulation of TRPA1 with AITC, inflammatory parameters were observed, revealing a significant reduction in levels of NO, IL-1β, IL-6, MCP-1, MIP-2, and NF-κB, which may serve as mediators of the anti-inflammatory effect of TRPA1 in macrophages. In the study of human coronary artery endothelial cells (HCAEC), it was found that TRPC3 may mediate Ca^2+^ influx through the CAM/CAMK signaling pathway and activate NF-κB, and via NFκB the control of VCAM-1 expression in HCAEC is involved in the pathogenesis of atherosclerosis. In addition, TRPC3 also affects monocyte adhesion in HCAEC [193]. In addition, TRPC3 is involved in the early and late pathological changes of atherosclerosis. Tano [194] transplanted the bone marrow of TPRC3/ApoE double knockout mice to ApoE knockout mice and confirmed that after phenotype transformation, they were given a high-fat diet for 3 weeks/8 weeks, and the area of aortic root lesion, neutral lipid content, and the number of macrophages decreased during 3 W examination; 8 W examination showed that the necrotic core area decreased by 40% and had higher collagen content and greater fiber cap thickness, but there was no significant difference between the lesion area, neutral lipid content and the number of macrophages.

The measurement of aortic atherosclerotic surface lesion size showed that the lesion area of TRPM2 knockout mice was smaller than that of WT mice. TRPM2 knockout can alleviate the development of aortic atherosclerotic plaque, and the related mechanism may be related to the reduction of the expression of various atherosclerosis-related proteins and the production of inflammatory cytokines and ROS by TRPM2 knockout [195]. The proliferation and migration of VSMC induced by ox LDL are related to the occurrence and development of atherosclerotic plaque. The increase in MEK/ERK phosphorylation caused by the ox LDL activation of TRPM7 may mediate the proliferation and migration of VSMC, suggesting that blocking TRPM7 may have potential in treating atherosclerosis [196].

### 3.4. TRP Channel and Dyslipidemia, NAFLD

Dyslipidemia pertains to the atypicality of lipid mass spectrometry, encompassing heightened levels of total cholesterol, triglycerides, and low-density lipoprotein cholesterol (LDL-C) in serum, as well as diminished levels of high-density lipoprotein cholesterol (HDL-C). The presence of dyslipidemia can increase the likelihood of developing cardiovascular disease, non-alcoholic fatty liver disease, and tumors. Furthermore, a correlation exists between dyslipidemia and NAFLD. There exists a correlation between the serum levels of various lipid mass spectrometry constituents, including but not limited to lipoprotein a, apolipoprotein, and lipoprotein lipase, which may contribute to the accumulation of fat in non-alcoholic fatty liver disease (NAFLD) by affecting lipid metabolism [197].

Non-alcoholic fatty liver disease (NAFLD) refers to fatty degeneration of more than 5% of liver cells, characterized by liver lipid accumulation, and may progress to cirrhosis and liver cancer [198]. There are many pathogeneses of NAFLD, among which the double strike model is widely accepted and popular. This model explains that the first hit of insulin resistance (IR) leads to liver lipid accumulation, and the second hit of lipotoxicity leads to mitochondrial damage, oxidative stress, and endoplasmic reticulum (ER) stress and other pathological results, triggering liver injury and apoptosis [199].

The calcium ion, a widely recognized second messenger, plays a crucial role in regulating the material metabolism and hepatocyte regeneration of the liver. The disruption of calcium ion homeostasis in the liver may result in metabolic and regeneration disorders, contributing to the development of non-alcoholic fatty liver disease (NAFLD). The cytoplasmic membrane of liver cells contains TRPV1, TRPV4, and TRPM2, which mediate the influx of extracellular calcium ions into the cytoplasm and regulate cellular calcium homeostasis. Alterations in the expression levels of these proteins may either inhibit or promote the progression of NAFLD [200]. The findings indicate an elevation in CYP2E1 activity in NAFLD. Within Kupffer cells, the TRPV4-eNOS pathway can generate NO, which diffuses in a paracrine manner to neighboring hepatocytes to impede CYP2E1-mediated redox toxicity and counteract tissue harm. CYP1E2-induced oxidative stress can trigger the upregulation of DNA methyltransferase I (Dnmt1), resulting in TRPV4 DNA promoter methylation and a significant decrease in TRPV4 expression. Consequently, CYP2E1-mediated redox toxicity contributes to liver inflammation and damage, thereby contributing to the development of NAFLD [201].

The activation of TRPC5 is involved in the development of cholestasis and related dyslipidemia. Alawi et al. [109] used bile acid to feed WT mice and TRPC5 knockout mice for 21 days to study the serum lipid mass spectrometry and liver gene expression of lipid homeostasis medium. The results showed that the liver triglycerides, total cholesterol, and LDL/VLDL levels of WT mice increased; the expressions of sterol regulatory element binding transcription factor 1 (Srebf1), fatty acid synthase (Fasn), stearoyl-CoA desaturase-1 (Scd1) and apolipoprotein E (ApoE) increased. The above changes in blood lipids and the expression of mediators did not change significantly in TRPC5 knockout mice.

Zhang et al. [202] used cholesterol-free dieting ovariectomized (OVX) rats as experimental objects, which were given capsaicin at doses of 0 mg/kg, 5 mg/kg, 10 mg/kg, and 15 mg/kg in groups. The lipid level detected showed that with the increase in dose, plasma TC and LDL-C decreased significantly, while liver TG decreased relatively insignificantly. At the same time, the study of gene expression in the liver showed that the mRNA levels of HMG CoA reductase and CYP7A1 decreased with the increase in capsaicin dosage, while the TRPV1 mRNA level increased. It is possible that capsaicin can participate in cholesterol metabolism by activating TRPV channels (such as by inhibiting the synthesis of cholesterol in the liver, stimulating the conversion of cholesterol into bile acid, or increasing the excretion of bile acid) to play its cholesterol-lowering role [203,204]. In addition, it is reported that the meta-analysis of a human control trial also showed that the supplementation of TRPV1 channel agonist capsaicin could reduce human serum total cholesterol and LDL levels through dietary intervention [205].

Capsaicin-mediated TRPV1 activation can eliminate lipid deposition in hepatocytes and liver tissues. Li et al. conducted an investigation into the mechanism of action of capsaicin and capsazepine on HepG2 cells treated with FFA. The authors employed a PPARδ agonist (GWo742) and antagonist (GSKo66o) and observed the effects on autophagy [206]. The results indicated that GWo742 augmented the capsaicin-induced increase in autophagy, while GSKo66o enhanced the inhibition of capsazepine-induced autophagy. Simultaneously, in vivo experimental findings demonstrated that capsaicin could enhance liver PPARδ expression in wild-type mice and autophagy, while exhibiting no significant impact on TRPV1 knockout mice. This suggests that TRPV1 is reliant on PPARδ to influence hepatocyte autophagy and contribute to liver lipid accumulation. Furthermore, the activation of TRPV1 by chronic dietary capsaicin has been shown to enhance autophagy via PPARδ, thereby serving as a preventative measure against non-alcoholic fatty liver disease. The upregulation of phosphorylation hormone-sensitive lipase (HSL), carnitine palmitoyl transferase 1 (CPT1), and peroxisome proliferator-activated receptor-δ (PPARδ) in the liver of wild-type mice following chronic dietary capsaicin consumption may also contribute to the breakdown of liver fat. Furthermore, the induction of uncoupling protein 2 (UPC2) expression in the liver of wild-type mice through the activation of TRPV1 by dietary capsaicin has been shown to enhance hepatic β-oxidation, modulate lipid metabolism, and mitigate non-alcoholic fatty liver disease (NAFLD) by decreasing hepatic lipid accumulation [207].

## 4. Conclusions and Prospect

Given the high prevalence of metabolic syndrome and its associated risk factors, including but not limited to obesity, insulin resistance, impaired glucose tolerance, hypertension, and dyslipidemia, which pose a significant threat to the development of diabetes, cardiovascular disease, and non-alcoholic fatty liver disease, it is imperative that we prioritize the study of metabolism-related diseases and develop novel approaches for their diagnosis and treatment. With the aim of elucidating the correlation between TRP channels and metabolic-related disorders, we conducted a comprehensive review of the expression and function of TRP channels in metabolic tissues, as well as their influence on certain metabolic diseases.

As previously stated, a diverse range of TRP channels have the capacity to modulate insulin secretion via the regulation of calcium influx and membrane depolarization through distinct signal transduction pathways. These channels include TRPA1, TRPM2, TRPV1, TRPV2, TRPV4, and TRPC1, among others. Additionally, TRPV1 may be involved in the regulation of pancreatic islets β-cell survival and apoptosis, sensory abnormalities in diabetic peripheral neuropathy, and microcirculation vascular dysfunction. Given our concern regarding the role of TRPV1 in insulin secretion, as well as its potential involvement in diabetes-related vascular lesions and neuropathy, we posit that TRPV1 represents a promising avenue for future research into the pathogenesis and treatment of diabetes. We eagerly anticipate further advancements in this area of inquiry. Various studies have indicated that distinct TRP channels may be involved in the pathogenesis of hypertension, including TRPM7 (leptin-induced hypertension), TRPV1 (cardiac sympathetic afferent reflex (CSAR) induced hypertension), TRPM5 (high-salt-induced hypertension), and TRPM8 (cold-induced hypertension). These findings suggest that targeted treatment of hypertension based on its underlying causes may yield superior therapeutic outcomes in the future. TRP channels in the liver are implicated in a variety of physiological and pathological processes, including liver regeneration (TRPM8), hepatocyte damage due to oxidative stress (e.g., TRPV4, TRPM2), liver ischemia and perfusion injury (e.g., TRPM2, TRPM6, TRPM7, TRPM8), liver fibrosis (TRPV1), and hepatocellular carcinoma (e.g., TRPC6, TRPV2), among others. These findings underscore the potential of TRP channels as a promising research target for metabolic-related diseases in related metabolic tissues. Furthermore, TRP channels exert regulatory effects on endothelial function through the modulation of vascular contraction and relaxation, inflammatory responses of endothelial cells, and phenotypic transformation and proliferation, as well as the migration of VSMC, thereby contributing to the pathogenesis of diabetes, hypertension, AS, and other related disorders. Additionally, certain redox-sensitive TRP channels (namely TRPM2, TRPM7, TRPC5, TRPV1, and TRPA1) can serve as sensors that activate cellular responses to oxidative stress, thereby regulating redox homeostasis. The involvement of ROS in the pathological damage process may lead to the development of diabetes, hypertension, and AS. The regulation of cholesterol metabolism by TRPV1, TRPA1, and TRPC5 through distinct signal pathways can impact lipid accumulation in cells, foam cell formation, and lipid deposition in liver tissue, thereby contributing to the development of AS and NAFLD.

The findings presented herein suggest that the Transient Receptor Potential (TRP) channel influences metabolism-related disorders not only through its expression and function in metabolic tissues and organs, but also by affecting the functions of both macro and micro blood vessels, oxidative stress, and lipid metabolism, among other factors (Figure 5). This suggests that systemic histopathological changes extending beyond metabolic tissues and organs should be considered. Further exploration and elucidation of the pathogenesis of TRP channels in metabolism-related diseases are essential for future research. Moreover, the translation of pertinent research outcomes into clinical applications is crucial to benefit patients suffering from such diseases. The development of TRP-targeted therapeutics faces primary challenges such as the specificity of TRP-ligand interactions and the sustainability of their therapeutic effects. This review critically assesses the existing research on the role of TRP in maintaining metabolic homeostasis in key organs including the pancreas, liver, and adipose tissue. It also considers the main difficulties in therapeutically targeting TRP channels for metabolic disorders and related complications. Investigations into TRP-associated pharmaceuticals are expected to address these challenges and offer valuable insights for the development of more effective and sustainable treatments. This systematic review is anticipated to significantly expedite the design of TRP-associated compounds for the improvement of metabolic disorders and related complications. This review highlights the regulatory role of TRP through associated pharmaceuticals and the modulation of TRP channels in the pancreas, liver, and adipose tissue, as well as the mechanisms of pathogenesis. These discussions may stimulate further discoveries of drugs targeting TRP channels for human use. It is hoped that this comprehensive summary will assist researchers from various disciplines in designing more effective ion channel-targeting drugs for the treatment of metabolic disorders, complications, and other chronic diseases.

## Figures and Tables

**Figure 1 ijms-25-00692-f001:**
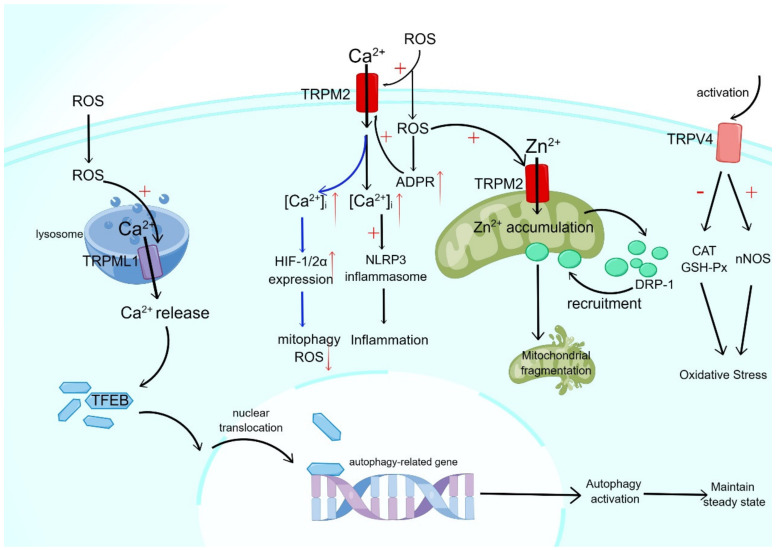
TRP channels and oxidative stress. Red upward arrows indicate activation or induction, red downward arrows indicate inhibition; "+" indicates up-regulation or enhancement, "−" indicates down-regulation or reduction. Redox-sensitive TRP channels can act as sensors to trigger cellular responses to oxidative stress stimuli (e.g., hydrogen peroxide, NO, electrophile reagents, etc.), including cell death, inflammatory responses, and cytokine release. The figure shows the signaling relationship between some TRP channels and oxidative stress; for example, the activation of TRPV4 can cause oxidative stress in cells, and ROS can activate TRPM2 in a direct or indirect way, leading to inflammatory response and mitochondrial fragmentation, and may also play a beneficial role by participating in affecting mitochondrial autophagy and ROS reduction. ROS can also activate TRPML1 on the lysosomal membrane to affect the nuclear translocation of TFEB and influence autophagy.

**Figure 2 ijms-25-00692-f002:**
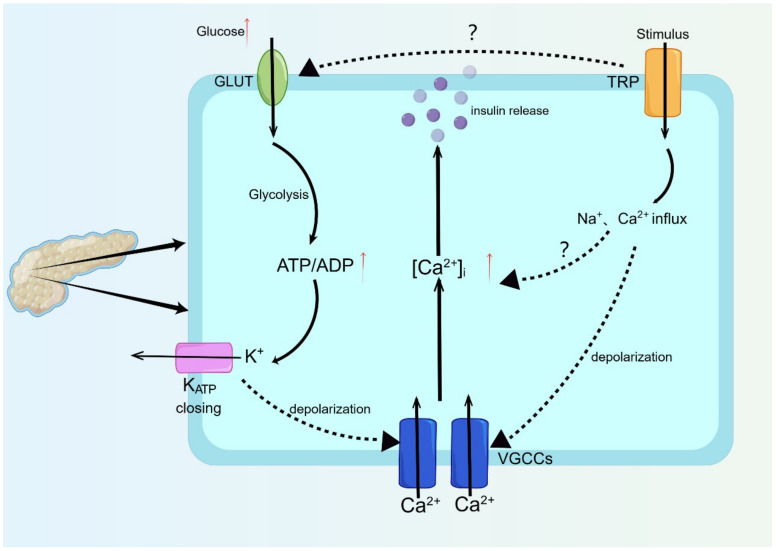
The mechanism of insulin secretion. Red upward arrows indicate up-regulation or increase, and dashed lines indicate potential mechanisms or accompanying changes in cell polarization state due to ion flow. "?" indicates potential mechanism. See text for details. As shown in the figure, the GSIS mechanism is when elevated plasma glucose enters the cell via GLUT and subsequently enters the glycolytic pathway for metabolism, accompanied by the consumption of ADP and the production of ATP, and the elevated ATP/ADP ratio causes K^+^_ATP_ closure and cell membrane depolarization; the plasma membrane depolarization makes the VGCCs open and causes calcium ion inward flow; and the increased intracellular calcium ion concentration triggers the calcium ion-dependent cytosolic release of insulin vesicles in β-cells. TRP channels, as a nonselective cation channel, can mediate non- K^+^_ATP_ -dependent Ca^2+^ inward flow and membrane depolarization in pancreatic β-cells to participate in insulin release. In addition, the role of TRP channels for GLUT and the increase in intracellular calcium ions due to TRP channels themselves may serve as potential mechanisms of insulin secretion.

**Figure 3 ijms-25-00692-f003:**
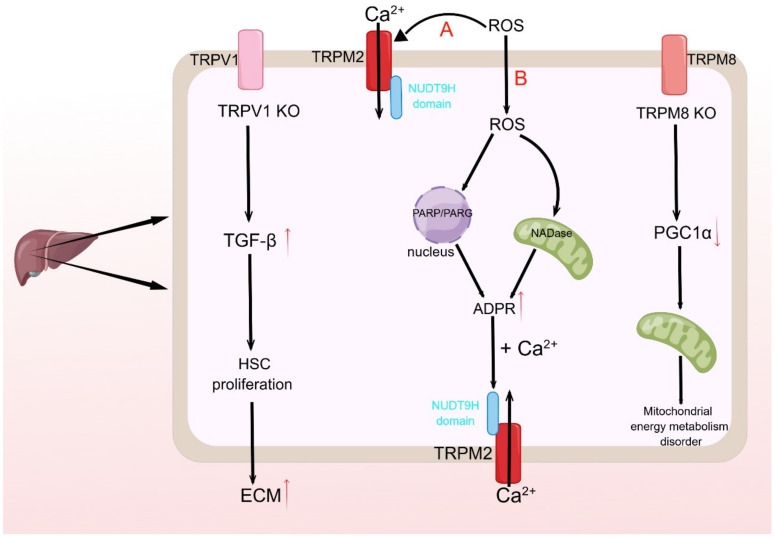
TRP channels and the liver. The red upward arrows in the figure indicate up-regulation, activation, or facilitation; conversely, the red downward arrows indicate down-regulation or inhibition. TRPV1 knockdown promotes TGF-β-induced HSC proliferation and activation, which in turn leads to increased ECM due to elevated extracellular matrix protein expression, which may be associated with liver fibrosis; TRPM8 knockdown can lead to impaired mitochondrial energy metabolism in hepatocytes. TRPM2 can be activated directly (pathway A in the figure) or indirectly (pathway B in the figure) by ROS. In the B pathway, ROS entry into cells can cause increased ADPR production by affecting PARP/PARG in the nucleus or NADase in mitochondria, and the increased ADPR can open the channel by binding to the NUDT9H structural domain of the TRPM2 channel together with Ca^2+^, mediating ROS-induced liver injury.

**Figure 4 ijms-25-00692-f004:**
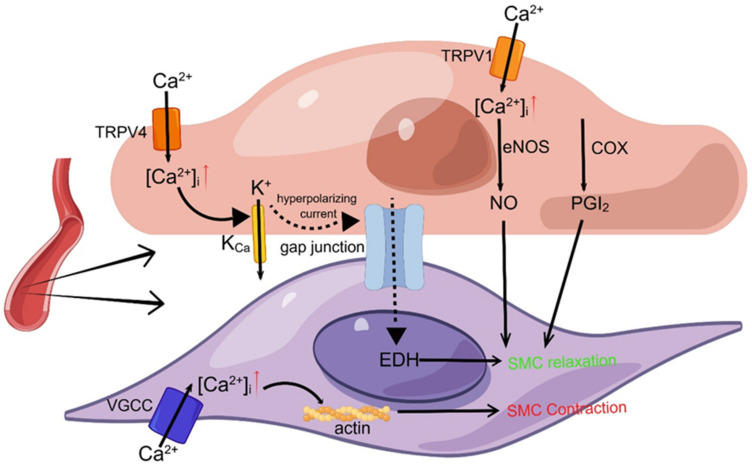
TRP channels and vasoconstriction and diastole. The figure shows calcium-mediated smooth muscle contraction in vascular smooth muscle cells and three endothelium-dependent vasodilatory pathways in endothelial cells: (1) eNOS-NO, (2) COX-PGI2, and (3) EDHF. The red upward arrow indicates an increase in ion concentration. TRPV1 causes calcium inward flow, and elevated intracellular calcium ions catalyze the production of a tissue-permeable gas NO via eNOS, which enters the SMC and causes vasodilation; calcium ion influx through a single TRPV4 channel forms a local calcium signal that activates calcium-sensitive potassium channels (e.g., SK, IK), whose mediated hyperpolarizing currents diffuse through gap junctions to surrounding smooth muscle cells to hyperpolarize them and cause vasodilation.

**Figure 5 ijms-25-00692-f005:**
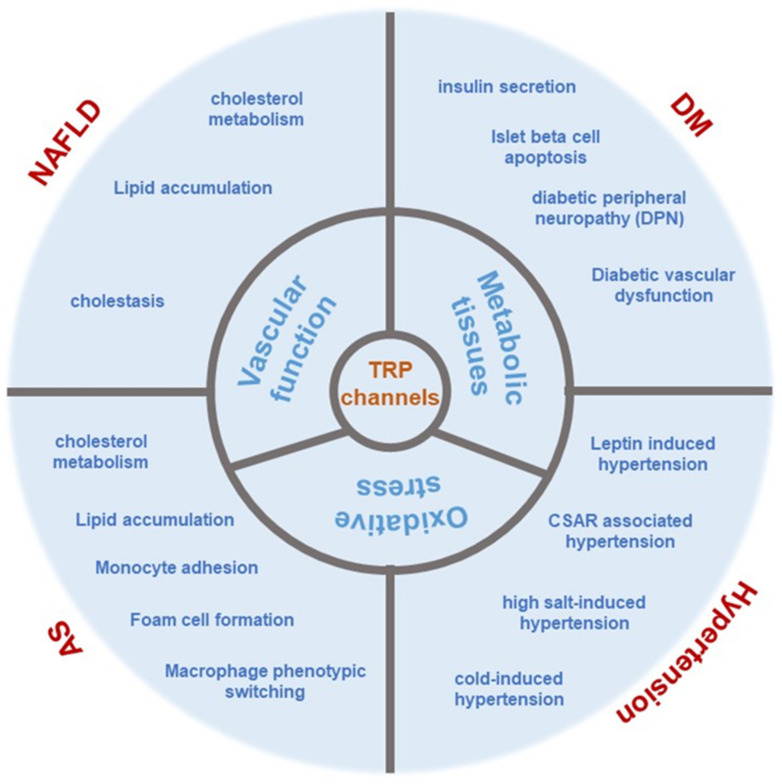
The relationship between TRP channels and metabolic-related diseases. The figure is divided into three levels: TRP channels, related pathophysiological processes and metabolic-related diseases, showing that TRP channels can participate in the development of metabolic-related diseases (e.g., DM, hypertension, AS, NAFLD, etc.) through their expression and function in metabolic tissues or organs, and their effects on vascular endothelial function and oxidative stress. It is noteworthy that there may also be crossover and interactions between the three layers and within each layer; for example, oxidative stress and vascular dysfunction affect metabolic tissues and organs, and pancreatic dysfunction or hepatic dysfunction may also be involved in the formation of the former two, and cholesterol metabolism and lipid accumulation jointly affect the development of AS and NAFLD. The core idea that the picture is intended to convey is the interaction between the three layers and within each layer, and further elaboration of their interactions is expected in the future.

## Data Availability

The datasets used and/or analyzed during the current study are available from the corresponding author on reasonable request.

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
