# Peer review of "Role of TRP Channels in Metabolism-Related Diseases"

_ijms, 2024, doi:10.3390/ijms25020692_

Round 1

Reviewer 1 Report

Comments and Suggestions for Authors

This review manuscript by F. Wu and coworkers compiles valuable information on
the importance of the different transient receptor potential channels in metabolic
diseases.
The manuscript highlights the importance of metabolic diseases, which are
constantly growing in developed countries, with high associated morbidity and
mortality. They also indicate the need to continue the research related to the role
of TRP channels in these diseases, as a way to develop new future drugs.

 The content is appropriate, they incorporate a few clarifying figures, which help in a better comprehension, and the references cover recent literature up to 2022. The heading and subheading seems also correct.

To complete this interesting review, the authors could comment and include some appealing 2023 references: i.e.  J Hypertension 2023, 41(9), 1351-70; ChemMedChem 2023, 18(4) e202200562, among others.

Regarding language, perhaps some editing could avoid unnecessary repetitions. For instance, in page 4, last paragraph, 7-9 lines from the bottom, there are two sentences that essentially say the same.

Other minor points:

Within the text, separate references from the preceding word.

Page 7, 2.4 heading: use channels (plural) instead channel (singular)

Reviewer 2 Report

Comments and Suggestions for Authors

This article aimed to review the current knowledge regarding the role of TRP channels in metabolism-related diseases. The paper is interesting. Nevertheless, the Authors should extend it and make more effort to study the literature. These are my points for the Authors:

1.      The information provided in Section 1. should be extended and summarized in the table. All members of TRP families should be listed. Furthermore, it would be worthwhile to add more data regarding their tissue distribution and exo- and endogenous modulators.

2.      Section 2.1( Function of TRP channels in the pancreas) focuses mainly on the role of selected TRP channels (but not all) in controlling insulin secretion. Therefore, the Authors should consider incorporating this part in section 3.1 (TRP channel and diabetes). Furthermore, some important papers regarding the role of TRP channels in controlling pancreatic beta cell functions should have been included. Therefore, I suggest studying the literature more deeply.

3.      It is disappointing that the Authors omitted the potential role of TRP channels in the biology of adipose tissue, adipogenesis, and obesity. The Authors should address this point in the new version of the manuscript. Adipose tissue is mentioned in the abstract as well as in the introduction. Overall, since this is a review article aiming to discuss the role of TRPs in metabolic disease, this point should be addressed, and the literature should be studied more thoroughly. Some chapters must be completed in the current form, and numerous essential papers were omitted.

4.     Typing errors such as double dots (e.g., the first sentence in section 11) exist. Therefore, the manuscript should be carefully revised.

Reviewer 3 Report

Comments and Suggestions for Authors

In this review article Wu et al. discuss the role of Transient receptor potential (TRP) channels in metabolism-related diseases.

The topic of the manuscript is interesting, but the article is only a descriptive and confused analysis of the studies reviewed.

In my opinion, the review does not provide a critical view of the authors. Thus, the Authors should discuss in a critical manner the main results emerging from the literature analysis, their strengths, and limitations. 

The manuscript needs substantial revision, as outlined below:

1. The abstract should better be structured as follows: a brief but precise presentation of the topic, a description of the research method, the most important results, and the conclusions.

In the present version of the abstract, the reader is only informed about the role of TRP channels at the end. In addition, the term “TRP” should be defined.

2. “Previous studies conducted from 2004 to 2014 have estimated that approximately 25% to 33% of adults from diverse ethnic backgrounds meet the MetS criteria”.

The time interval ends in 2014 and is not current.

3. The introduction should be better structured.

Authors should begin with an overview of the topic and provide a context explaining why a review of the topic is necessary. How is the following organization into sections?

As it stands, the introduction seems to consist of two paragraphs with no obvious connection between them.

4. The seven families of TRPs should be better described in a table, which also includes the main characteristics of each receptor. 

5. The title “2. TRP channel and organ tissue, oxidative stress” is not clear.

In my opinion, the paragraph should be restructured by first discussing the role of TRPs in oxidative stress and then their localization at the level of different organs/tissues and their specific function in molecular processes related to metabolic diseases.

6. Paragraph 3 is also not logically structured.

In my opinion, attention should be paid to the involvement of TRPs in the biochemical/physiological processes underlying the pathologies described.

7. The authors should critically discuss the reported data and draw a clear conclusion with possible implications for the treatment/prevention of metabolic diseases.

Round 2

Reviewer 2 Report

Comments and Suggestions for Authors

I have no more no comments for the Authors.

Reviewer 3 Report

Comments and Suggestions for Authors

The Authors have addressed all my concerns and I have no further comments. 

As far as I am concerned, the manuscript is now acceptable to be published.